# Peripheral Blood Mononuclear Cell Populations Correlate with Outcome in Patients with Metastatic Breast Cancer

**DOI:** 10.3390/cells11101639

**Published:** 2022-05-13

**Authors:** Anna-Maria Larsson, Olle Nordström, Alexandra Johansson, Lisa Rydén, Karin Leandersson, Caroline Bergenfelz

**Affiliations:** 1Division of Oncology, Department of Clinical Sciences Lund, Lund University, SE-223 81 Lund, Sweden; anna-maria.larsson@med.lu.se (A.-M.L.); al0531jo-s@student.lu.se (A.J.); lisa.ryden@med.lu.se (L.R.); 2Experimental Infection Medicine, Department of Translational Medicine, Lund University, SE-214 28 Malmö, Sweden; ol7840no-s@student.lu.se; 3Department of Surgery, Skåne University Hospital, SE-223 81 Lund, Sweden; 4Cancer Immunology, Department of Translational Medicine, Lund University, SE-214 28 Malmö, Sweden; karin.leandersson@med.lu.se

**Keywords:** metastatic breast cancer, peripheral blood, lymphocytes, myeloid cells, prognosis

## Abstract

Local tumor-associated immune cells hold prognostic and predictive value in various forms of malignancy. The role of systemic, circulating leukocytes is, however, not well-characterized. In this prospective and explorative study, we aim to delineate the clinical relevance of a broad panel of circulating immune cells in 32 patients with newly diagnosed metastatic breast cancer (MBC) before the start of systemic treatment. Freshly isolated peripheral blood mononuclear cells (PBMCs) were analyzed by flow cytometry and evaluated for potential associations to clinicopathological variables and patient outcome. We show that the levels of specific circulating leukocyte populations are associated with clinical parameters such as hormone receptor status, histological subtype, number of circulating tumor cells (CTCs) and metastatic burden. Importantly, high levels of CD8^+^ cytotoxic T lymphocytes (CTLs) are significantly linked to improved overall survival (OS). In patients with estrogen receptor (ER)-positive primary tumors, high levels of circulating CTLs and non-classical (CD14^+^CD16^++^) monocytes were associated with improved OS, whereas in patients with ER-negative tumors low levels of circulating natural killer (NK) cells potentially associate with improved OS. We propose that the levels of specific circulating immune cell populations, such as CD8^+^ CTLs, may be used to predict clinical outcomes in MBC patients. Thus, larger studies are warranted to validate these findings.

## 1. Introduction

Breast cancer is the most common malignancy in women worldwide [1]. Due to improvements in early diagnosis and adjuvant therapy, early-stage localized breast cancer is considered curable in approximately 70–80% of patients [2], while 20–30% of the patients will develop metastatic disease with dissemination to distant organs [1,2]. Metastatic breast cancer (MBC) is treatable, but in most cases not curable, and accounted for more than 500,000 deaths globally in 2015 [1]. With the emerging use of new targeted therapies, there is an urgent need to improve current, and identify novel, prognostic and treatment predictive markers to personalize treatment strategies in MBC patients [3].

Over the past decades, the role of the immune microenvironment in breast cancer development and progression has become increasingly recognized [2,4]. High numbers of tumor-infiltrating lymphocytes (T-, B-, natural killer [NK] and NKT cells) are generally associated with more favorable clinical outcomes in breast cancer, especially in patients with HER2-positive and triple-negative tumors [5,6,7,8,9,10,11,12] but also in estrogen receptor (ER) positive disease [13]. In contrast, tumor-infiltrating myeloid cells (e.g., tumor-associated macrophages [TAMs] and dendritic cells [DCs]) are generally associated with poor prognosis [14,15,16,17]. The clinical significance of systemic, circulating immune cell subpopulations is, however, less clear.

As of today, only a handful of studies have examined the association of circulating immune cell populations with clinical parameters and/or outcomes in breast cancer patients, especially in MBC patients [18]. Furthermore, the majority of studies utilize cryopreserved peripheral blood mononuclear cells (PBMCs), which may affect viability, as well as surface marker expression, on PBMCs [19,20,21]. In two recent studies, we report that peripheral blood monocytic myeloid-derived suppressor cells (Mo-MDSCs; CD14^+^HLA-DR^low/−^ cells) are enriched in MBC patients compared to healthy donors and that high levels of Mo-MDSCs correlate with ER-negativity, metastatic disease at the initial breast cancer diagnosis (de novo MBC) and with worse outcome [22,23]. Similarly, in patients with HER2-positive MBC, high levels of circulating CD4^+^ and CD3^+^ T lymphocytes are associated with worse survival (progression-free and overall survival, respectively) [24]. In addition, a low monocyte-to-lymphocyte ratio was associated with improved overall survival in MBC patients, particularly in patients with hormone receptor and HER2-negative (triple-negative) breast cancer [25]. A high frequency of TNFα-producing myeloid dendritic cells (MDCs), but not total DC, MDC or plasmacytoid dendritic cell (PDC) levels, associated with worse overall survival in patients with locally advanced or metastatic inflammatory breast cancer [26]. In MBC patients receiving high-dose chemotherapy, high frequencies of mature CD14^+^HLA-DR^+^ monocytes, as well as specific subpopulations of CD8^+^ and CD4^+^ T lymphocytes, are associated with longer breast cancer-specific survival [27]. We have previously shown that systemic treatment, including chemotherapy and endocrine therapy, affect the levels of circulating immune cell populations [28]. Whether the frequencies of specific circulating immune cell populations at baseline, before start of systemic treatment, hold prognostic or treatment predictive value is not well-characterized in MBC patients.

In this prospective observational study, we explore the systemic immune response by analyzing circulating mononuclear immune cell populations from 32 patients with newly diagnosed MBC, before the start of systemic treatment, with the aim of assessing whether specific immune cell populations correlate with clinical parameters or outcome. To our knowledge, this is the first study to comprehensively characterize the clinical relevance of a broad panel of lymphocyte and myeloid cell populations in fresh blood (not cryopreserved) from MBC patients in association with clinical parameters and patient outcome.

## 2. Materials and Methods

### 2.1. Patients

This is a sub-study of the prospective observational CTC-MBC trial (ClinicalTrials.gov NCT01322893) and clinical information regarding the CTC-MBC study has previously been published [29]. Thirty-two patients with newly diagnosed MBC were included in this study (see Appendix A for clinical information regarding the specific patients included in the current study). Data regarding the median percentages of monocytes, T lymphocytes, B lymphocytes, NK cells, NKT cells, MDC1, MDC2 and PDCs from the first 23 patients have previously been published in relation to median percentages from healthy donors, but no associations were made to clinical parameters [22].

Circulating tumor cells (CTCs) were detected as previously described [29]. Briefly, CTCs were enumerated in blood samples within 96 h of collection using the CellSearch system (Menarini Silicon Biosystems, Bologna, Italy) and the established cut-off at five CTCs was used for dichotomization into high/low groups [29]. Data regarding all CD14^+^ monocytes, not separating in CD16^+/−^subpopulations, as well as CD14^+^HLA-DR^low/−^ Mo-MDSCs from these patients, have previously been analyzed in relation to heathy donors and to clinical parameters [23].

An ethical permit was obtained from the Research Ethics Committee at Lund University (Dnr 2010/135, Dnr 2010/477 and Dnr 2011/748). All patients signed a written informed consent, and the study was performed in accordance with good clinical practice (GCP) and the Declaration of Helsinki. The primary endpoint in the original study was progression-free survival (PFS) and overall survival (OS) was a secondary endpoint.

### 2.2. Cell Isolation 

Peripheral blood samples were drawn in EDTA-coated tubes prior to the start of systemic therapy. All blood samples were analyzed within 24 h. The samples were diluted 1:2 in sterile phosphate buffered saline (PBS) supplemented with 2mm EDTA and 2.5% (*w*/*v*) sucrose [PBS/EDTA/sucrose], overlaid on Ficoll-Paque Plus (GE Healthcare, Uppsala, Sweden) and centrifuged with the brakes off at 400× *g* for 30 min at room temperature. Peripheral blood mononuclear cells (PBMCs) were collected and washed in PBS/EDTA/sucrose at 350× *g*, 4 °C for 7 min.

### 2.3. Flow Cytometry

From 10,000–50,000 cells were subsequently stained for flow cytometry for a total of 20–30 min at 4 °C in a FACS buffer (PBS supplemented with 3% fetal bovine serum and 0.1% sodium azide). All antibodies have been used in previously published studies [22,28] and were carefully evaluated and titrated before use. Antibodies used (with clones and dilutions) are specified in Appendix A. Due to variable sample volumes, we were not able to perform all analyses on all patients (see Appendix A). All analyses were performed using FACS Calibur (BD Biosciences, San Jose, CA, USA) and on gated viable (as assessed by 7AAD-negativity) PBMCs (excluding contaminating granulocytes) using the Cell-Quest pro software (version 6.0, BD).

### 2.4. Statistical Analyses

All statistical analyses were performed using SPSS Statistics (version 27, IBM, Armonk, NY, USA) and STATA (version 16.1, StataCorp, College Station, TX, USA). Patients were dichotomized into either “high” or “low” groups, based on the median level (i.e., the percentage) of indicated immune cell populations (Appendix A). Pearson’s chi-square test or Fisher’s exact test (when the expected counts were less than five in at least one cell) were used to compare categorical patient and tumor characteristics in relation to “high” or “low” levels of immune cells. Progression-free (PFS) and overall survival (OS) were analyzed by log-rank test and visualized by Kaplan-Meier curves. In addition, for CD8^+^ CTLs, uni- and multivariable Cox regression analyses (adjusting for age, ECOG status, Nottingham histological grade (NHG), primary tumor histological subtype, metastasis-free interval (MFI), number of metastases and presence of visceral metastasis) were performed. *p*-values < 0.05 were considered significant. 

## 3. Results

### 3.1. Patient Characteristics

This is a sub-study of a larger prospective study comprising 156 patients with newly diagnosed MBC. Clinical information regarding this cohort has previously been published [29]. Immune cell populations from 32 patients with newly diagnosed MBC were analyzed in this prospective and explorative study. The sub-cohort is representative of the full cohort and shows no significant discrepancies with regards to baseline clinicopathological variables (age, ECOG, NHG, subtype, no. of metastatic loci, metastasis-free interval, visceral metastasis, number of CTCs and CTC-clusters; data not shown). The median age at MBC diagnosis of the included patients in the sub-cohort was 63 years (SEM ± 1.8 years, range 43–84 years, Appendix A). Regarding the immunohistochemistry (IHC) status of the primary tumor, 20 patients (62.5%) had estrogen receptor (ER)-positive primary tumors, whereas eight (25%) had ER-negative primary tumors. Five patients (15.6%) had HER2-positive primary tumors and four (12.5%) had triple-negative primary tumors (Appendix A). At MBC diagnosis, 13 patients (40.6%) had three or more metastatic sites. Twenty-five patients (78.1%) had bone metastases, 19 patients (59.4%) had visceral involvement, 12 patients (37.5%) had lymph node metastases, 12 patients (37.5%) had lung metastases and 9 (28.1%) had liver metastases (Appendix A). Five patients (15.6%) had de novo MBC at initial diagnosis, whereas 27 (84.4%) were diagnosed with distant recurrent MBC. The median metastasis-free interval for the included patients in the sub-cohort was five years (SEM ± 1.2 years). The median follow-up time was 54 months (range 49–83).

### 3.2. Associations between the Levels of Circulating T Lymphocyte Populations and Clinicopathological Variables

In a previous study, comprising the first 23 patients in the present study, we found that MBC patients have significantly lower levels of peripheral blood CD3^+^ T lymphocytes, NK cells, NKT cells, MDC1 and MDC2 compared to healthy donors [22]. Expanding on these findings in a larger cohort, we here investigated whether the levels of systemic, circulating immune cell populations associate with clinical parameters or survival. All peripheral blood samples were drawn before starting systemic therapy and the frequencies of peripheral blood mononuclear cell (PBMC) populations were analyzed by flow cytometry (see Figure 1 for gating strategies and Appendix A for antibodies used). Due to variable sample volumes, we were not able to perform all analyses on all patients (see Appendix A). The patient groups were then dichotomized into “high” or “low” based on the median percentage of indicated immune cell population (Appendix A). To determine the clinical relevance of circulating leukocyte populations, we first compared clinicopathological variables in patients with “high” versus “low” levels of specified leukocyte population (Appendix A). 

For all CD3^+^ T lymphocytes and CD4^+^ T-helper (Th) lymphocytes, there were no significant associations between patients with high and low levels regarding clinicopathological variables such as age, tumor histological subtype, Nottingham histological grade, tumor size, hormone receptor status, number or localization of metastases or type of MBC (MBC at initial breast cancer diagnosis [de novo MBC] or distant recurrent MBC, Appendix A). However, a significant correlation between high levels of CD8^+^ cytotoxic T lymphocytes (CTLs) and ER- and progesterone-receptor (PR) status in the primary tumor was observed (*p* = 0.033 and *p* = 0.001, respectively, Table 1 and Appendix A). Furthermore, low CTL levels correlated significantly to death (Table 1 and Appendix A). The levels of circulating regulatory T cells (T_regs_) associated with primary tumor histological subtype (*p* = 0.049) and Nottingham histological grade (*p* = 0.021, Table 1 and Appendix A). When analyzing the levels of CD8^+^ CTLs and CD4^+^ Th lymphocytes within all CD3^+^ T lymphocytes, high levels of CD8^+^ CTLs correlated with age (<65y, *p* = 0.013), positive PR and HER2 status of the primary tumor (*p* = 0.004 and *p* = 0.045, respectively), as well as with a type of MBC (de novo MBC, *p* = 0.043, Appendix A). High levels of CD4^+^ Th lymphocytes within all CD3^+^ T lymphocytes associated with negative PR status of the primary tumor, as well as of the metastasis (*p* = 0.00005 and *p* = 0.047, respectively, Table 1 and Appendix A).

### 3.3. Associations between the Levels of B Lymphocytes, NK or NKT Cells and Clinicopathological Variables

When analyzing patients with high or low levels of circulating B lymphocytes, high levels of CD19^+^ B lymphocytes associated with metastatic burden (≥3 metastatic sites, *p* = 0.047) and higher levels of circulating tumor cells (CTCs, ≥5 CTCs at baseline, *p* = 0.047, Table 1 and Appendix A). The levels of CD56^+^CD3^−^ NK cells associated with primary tumor histological subtype and PR status (*p* = 0.050 and *p* = 0.038, respectively), whereas there were no significant associations between patients with high and low levels of CD56^+^CD3^+^ NKT cells regarding the analyzed clinical variables (Appendix A).

### 3.4. Associations between the Levels of Monocyte Populations and Clinicopathological Variables

According to our previous study, circulating Mo-MDSCs, but not all CD14^+^ monocytes, associate with specific clinical parameters and with worse outcome [22,23]. However, as MBC patients tend to have higher levels of CD16^+^ monocyte populations (intermediate and non-classical monocytes) compared to healthy donors [22], we here analyzed the association between monocyte subpopulations (CD14^++^CD16^−^ classical monocytes, CD14^++^CD16^+/++^ intermediate monocytes and CD14^+^CD16^++^ non-classical monocytes) and clinical parameters, something that was not previously done. An inverse association between the levels of classical monocytes and primary tumor PR status was observed (*p* = 0.006), whereas the levels of non-classical monocytes associated with performance status (Eastern Cooperative Oncology Group [ECOG] status, *p* = 0.023, Table 1 and Appendix A). No other significant difference was observed between patients with high or low levels of these monocyte populations regarding clinical parameters. However, patients with high levels of intermediate monocytes tended to have more CTCs (≥ 5 CTCs at baseline, *p* = 0.034, Table 1 and Appendix A).

### 3.5. Associations between the Levels of Peripheral Blood Dendritic Cell Populations and Clinicopathological Variables

When analyzing the potential associations between the levels of circulating dendritic cell populations (BDCA-1^+^ MDC1, BDCA-3^+^ MDC2 and BDCA-2^+^ PDCs) with clinical parameters, there were no significant differences between patients with high or low levels of MDC2 and PDC for either clinical parameter investigated. High levels of circulating MDC1, however, associated with primary tumor histological subtype (higher prevalence of lobular breast cancer, *p* = 0.007), whereas low levels of MDC1 correlated with visceral metastases (*p* = 0.038, Table 1 and Appendix A).

### 3.6. High Levels of Circulating CTLs Associate with Improved Overall Survival

To investigate the prognostic impact of the analyzed circulating immune cell populations, we used Kaplan–Meier curves to visualize and a log-rank test to compare progression-free survival (PFS) and overall survival (OS) in patients with high or low levels of indicated immune cell population (Figure 2, Figure 3 and Figure 4 and Appendix A).

The 16 patients with high levels of CD8^+^ CTLs had a tendency toward improved PFS (*p* = 0.068, estimated median PFS for patients with high and low levels of CTLs; 20.9 months 95% CI 16.7–25.2 and 10.2 months 95% CI 0.0–29.8, respectively, Figure 2B left panel). Regarding OS, patients with high levels of CD8^+^ CTLs had a significantly improved OS compared to patients with low levels of CTLs (*p* = 0.003, estimated median OS for patients with high and low levels of CTLs; 54.2 months 95% CI 36.2–72.2 and 28.1 months 95% CI 7.3–48.8, respectively, Figure 2B right panel). A non-significant association was also observed between high levels of all T lymphocytes (CD3^+^ cells) and improved OS (*p* = 0.074, estimated median OS for patients with high and low levels of T lymphocytes; 44.2 months 95% CI 39.8–48.6 and 19.8 months 95% CI 0.0–65.6, respectively, Figure 2A right panel). No other significant associations to PFS nor OS were observed for patients with high or low levels of CD4^+^ Th lymphocytes, T_regs_, CD4^+^/CD8^+^ T cell ratio, B lymphocytes, NK cells or NKT cells (Figure 2 and Figure 3 and Appendix A).

### 3.7. Low levels of Intermediate Monocytes Associate with Improved Progression-Free Survival

When analyzing different monocyte subpopulations, we found that low levels of intermediate monocytes (CD14^++^CD16^+/++^, *N* = 13) associated with improved PFS (*p* = 0.041, estimated median PFS for patients with low and high levels of intermediate monocytes; 24.9 months 95% CI 11.6–38.3 and 8.5 months 95% CI 0.9–16.2, respectively, Figure 4B left panel), whereas there was no significant association to OS (Figure 4B right panel). A non-significant trend was also observed for high levels of non-classical monocytes (CD14^+^CD16^++^, *N* = 14) and improved PFS (*p* = 0.054, estimated median PFS for patients with high and low levels of non-classical monocytes; 21.2 months 95% CI 12.7–29.6 and 5.6 months 95% CI 3.9–7.4, respectively, Figure 4C left panel), as well as improved OS (*p* = 0.060, estimated median OS for patients with high and low levels of non-classical monocytes; 44.2 months 95% CI 39.8–48.6 and 19.8 months 95% CI 0.0–65.6, respectively, Figure 4C right panel).

### 3.8. High Levels of MDC2 Associate with Improved Progression-Free Survival

When analyzing different dendritic cell populations, we found that high levels of circulating MDC2 associated significantly with improved PFS (*p* = 0.050, estimated median PFS for patients with high and low levels of MDC2; 22.3 months 95% CI 14.4–30.1 and 10.8 months 95% CI 4.6–17.1, respectively, Appendix A left panel), but not to OS (*p* = 0.526, Appendix A right panel). No other trends were observed for patients with high or low levels of MDC1 or PDC regarding associations to PFS or OS (Appendix A).

### 3.9. High Levels of CTLs Associate with Improved Survival in Patients with ER-Positive Primary Tumors

To explore if the prognostic impact of different circulating immune cell populations differs in the ER-positive versus ER-negative disease, we analyzed PFS and OS in relation to the levels of circulating immune cells in patients stratified for the ER status of the primary tumors, hence some subpopulations in further analyses are very few (Figure 5 and Figure 6 and Appendix A). High levels of CTLs associated significantly with improved PFS (*p* = 0.048, estimated median PFS in patients with high and low levels of CTLs; 29.9 months 95% CI 19.2–40.7 and 21.1 months 95% CI 0.0–60.8, respectively, Figure 5B left panel), as well as OS (*p* = 0.015, estimated median OS in patients with high and low levels of CTLs; 59.9 months 95% CI 40.8–79.0 and 30.4 months 95% CI 0.00–64.8, respectively, Figure 5B right panel) in patients with ER-positive primary tumors.

### 3.10. Low Levels of NK Cells Potentially Associate with Improved OS in Patients with ER-Negative Primary Tumors

Regarding other lymphocyte populations, low levels of systemic NK cells were not associated with PFS but significantly associated with improved OS in patients with ER-negative primary tumors (*p* = 0.034, estimated median OS for patients with low and high levels of NK cells; 40.9 months [no 95% CI was obtained due to limited number of patients in this group] and 12.6 months 95% CI 0.48–24.6, respectively, Figure 5C right panel). However, these results need to be interpreted with caution because of the few patients in the analyses.

### 3.11. High Levels of Non-Classical Monocytes Associate with Improved Survival

When analyzing different monocyte populations in patients stratified for ER status, high levels of non-classical monocytes was significantly associated with PFS in patients with ER-negative primary tumors (*p* = 0.010, estimated median PFS in patients with high and low levels of non-classical monocytes; 10.2 months 95% CI 2.9–17.5 and 4.5 months 95% CI 0.0–10.7, respectively, Figure 6C left panel). In addition, a trend toward improved PFS in patients with ER-positive disease was observed (*p* = 0.071, estimated median PFS in patients with high and low levels of non-classical monocytes; 30.8 months 95% CI 15.6–46.0 and 5.7 months 95% CI 0.0–15.1, respectively, Figure 6C left panel). Furthermore, there was a significant association to improved OS for patients with high levels of non-classical monocytes and ER-positive primary tumors (*p* = 0.044, estimated median OS in patients with high and low levels of non-classical monocytes, not available because of too few patients and 30.4 months 95% CI 0.00–66.6, respectively, Figure 6C right panel).

### 3.12. Peripheral Blood Dendritic Cells Are Not Prognostic for Survival in Breast Cancer Patients

Regarding dendritic cell populations, there were no associations to PFS and only a borderline significant association observed between high levels of MDC1 and improved OS in patients with ER-positive tumors (*p* = 0.056, estimated OS for patients with high and low levels of MDC1; 59.9 months 95% CI 30.9–88.9 and 40.3 months 95% CI 12.0–68.7, respectively, Appendix A right panel).

### 3.13. Uni- and Multivariable Cox Regression Analyses of Survival in Relation to CTLs

Finally, we performed uni- and multivariable Cox regression analyses of survival in relation to CD8^+^ CTLs in the whole cohort, adjusting for age, ECOG, NHG, primary tumor histological subtype, metastasis-free interval, number of metastatic sites and presence of visceral metastasis. Similar to the log-rank analyses in Figure 2B, patients with high levels of CTLs displayed non-significant associations to improved PFS (HR_PFS_ 0.50, 95% confidence interval [CI] 0.023–1.07, *p* = 0.074 in univariable and HR_PFS_ 0.32, 95% CI 0.049–2.081, *p* = 0.23 in multivariable analyses). With regards to OS, patients with high levels of CTLs significantly associated with survival in univariable Cox regression analyses (HR_OS_ 0.28, 95% CI 0.12–0.69, *p* = 0.005). This significance was, however, lost when adjusting for other prognostic factors (HR_OS_ 0.51, 95% CI 0.066–3.98, *p* = 0.52).

## 4. Discussion

In this prospective and explorative study, we set out to expand on our previous findings regarding circulating immune cell populations in MBC patients [22,23], to now comprehensively evaluating a broad panel of circulating PBMC populations with regards to clinical impact. We found that patients with high levels of circulating CD8^+^ CTLs had an improved OS, and that this association was also found in patients with ER-positive disease. Furthermore, high levels of non-classical (CD14^+^CD16^++^) monocytes were associated with improved OS in patients with ER-positive primary tumors, whereas in patients with ER-negative tumors, low levels of circulating NK cells potentially associated with improved OS. Our data should also be compared to our initial study comprising the first included 23 patients [22], where we found that MBC patients display lower levels of peripheral blood CD3^+^ T lymphocytes, NK cells and NKT cells compared to healthy donors, whereas the levels of B lymphocytes and the percentages of CD8^+^ CTLs or CD4^+^ T-helper cells of all CD3^+^ T lymphocytes, as well as T_regs_ of all CD4^+^ cells, were not significantly different between MBC patients and healthy donors [22].

For MBC, the overall prognosis is poor and current treatment options are mainly directed to improve quality of life, prevent or alleviate symptoms and prolong survival [30]. Hence, there is an urgent need to identify novel targets, as well as prognostic and treatment predictive markers, to personalize treatment strategies for MBC patients. It has become increasingly recognized that tumors induce not only local, but also systemic immune changes. As of today, only limited information is available regarding the immune landscape in the peripheral blood of breast cancer patients and how it relates to clinical parameters and survival [18]. Most studies analyze one or a few immune cell populations from non-MBC patients or do not stratify based on presence or absence of metastases [11,12,26,31,32,33]. Furthermore, samples are often cryopreserved or obtained from patients under current systemic treatment [11,12,27].

Our findings show that high levels of CD8^+^ CTLs associated with hormone-receptor-positive primary tumors and a significantly improved PFS and OS in all MBC patients, specifically in patients with ER-positive primary tumors, a finding that was neither seen for the total CD3^+^ T lymphocyte population, nor for CD4^+^ T-helper cells or T_regs_. Although the significance was lost when adjusting for other prognostic factors, this warrants analysis in a larger cohort. Our results are also partially in line with a recent study reporting that high levels of distinct subpopulations of systemic CD8^+^ and CD4^+^ T lymphocytes associated with longer breast cancer-specific survival in MBC patients treated with high-dose chemotherapy [27]. Similarly, in inflammatory breast cancer, low levels of peripheral blood CD8^+^ and CD4^+^ T lymphocytes associate with inferior survival [33]. Our data is also in line with the reported association between high levels of tumor-infiltrating CD8^+^ T lymphocytes and improved survival in patients with early breast cancer [34,35,36]. CD8^+^ CTLs are considered important players in anti-tumor immune responses that recognize tumor antigens and mediate tumor-killing effects via, e.g., pro-inflammatory interferon γ (IFNγ) and cytotoxic granzyme or perforin [4,37]. Although this is considered a local response, our data indicate that high levels of circulating CD8^+^ CTLs may be a marker for improved outcome specifically in MBC patients with ER-positive primary tumors. It is possible that patients displaying high levels of CD8^+^ T lymphocytes in the peripheral blood also have a higher density of tumor-infiltrating CD8^+^ T lymphocytes. However, MDSCs are well-known suppressors of T lymphocyte function and an inverse association between systemic Mo-MDSCs and CD3^+^ T lymphocytes in MBC patients has previously been described [22]. Thus, the potential correlation between systemic and tumor-associated CD8^+^ T lymphocytes will be important to study in the future and especially in the setting of immune checkpoint inhibitors. Based on our findings, we will in future studies evaluate the presence of different immune cell populations in primary tumors compared to metastatic tissue.

Due to study setup and technical limitations from the lack of modern flow cytometers (resulting in the use of separate staining panels), expression of immune checkpoint markers such as PD-1 were not investigated on T lymphocytes. Neither were more in-depth analyses of T lymphocyte subpopulations, e.g., double negative and double positive T cells, different CD8^+^ CTL subsets nor CD4^+^ T lymphocyte cell subsets (including Th1, Th2, Th9 or Th17) technically feasible. The latter may explain why we did not observe any association between systemic CD4^+^ Th lymphocytes and any clinicopathological variable or survival, as has been proposed for tumor-associated T-helper subsets [6,38]. Furthermore, Yang et al. reported that low levels of CD3^+^ and CD4^+^ T lymphocytes, but not CD8^+^ T lymphocytes, associated with improved PFS and OS in MBC patients with HER2-positive but not luminal A, luminal B or triple negative tumors [24]. In our study only five of the 32 included patients had HER2-positive primary tumors; hence, no subgroup analysis of these patients could be presented. Tumor-infiltrating T_regs_ have previously been shown to correlate to worse prognosis, especially in patients with hormone-receptor-positive early breast cancer [38,39,40]. Our data regarding systemic T_regs_ indicate no association with either clinical parameters or outcome. This would be in accordance with data from patients with early, primary breast cancer, where circulating T_regs_ did not associate with patient survival, despite higher levels in breast cancer patients than healthy donors [31], implying that tumor-associated T_regs_ might be better prognostic indicators than systemic T_regs_. When analyzing systemic B lymphocytes, we found associations between high levels of B lymphocytes and metastatic burden (>3 metastatic sites), as well as high CTCs, yet no association with survival. This is in contrast to tumor-infiltrating B lymphocytes that have generally been reported to associate with favorable outcome, although this is still under debate [8,9,10]. Furthermore, in our previous study we found no difference in the levels of B lymphocytes between MBC patients and healthy controls [22] and current literature regarding systemic B lymphocytes is conflicting [18], warranting further studies of the role of circulating B lymphocytes in breast cancer patients.

NK cells recognize and kill tumor cells by, e.g., pro-inflammatory IFNγ and tumor necrosis factor alpha (TNFα), Fas-FasL signaling and cytotoxic granzymes and perforins [4]. As for CD8^+^ CTLs, this effect is considered local and a high density of tumor-associated NK cells associate with improved clinical outcome in breast cancer [11,12]. In a previous study, we noted that the levels of circulating NK cells were significantly lower in MBC patients compared to healthy donors [22]. Here we found that low levels of circulating NK cells potentially associated with improved OS in MBC patients with ER-negative primary tumors. However, these results need to be interpreted with caution because of the few patients in the analyses. Our data are in contrast to a previous study indicating an association between high levels of systemic NK cells and improved pathological response after neoadjuvant treatment in early breast cancer [12]. However, in that study, the patients were not stratified according to the ER status of the primary tumor and only mature CD16^+^CD56^dim^ NK cells were investigated. Furthermore, the levels of tumor-associated and peripheral blood NK cells have previously been shown to correlate inversely in patients with HER2-positive early breast cancer [11]. Although that study reported reduced levels of NK cells in primary breast tumors as compared with patient-matched peripheral blood, it is possible that the levels of NK cells are higher in ER-negative tumors as compared with patient-matched peripheral blood in patients with MBC and this warrants further studies. Patients with ER-negative primary tumors, and especially triple-negative breast cancer, generally have a worse prognosis as compared to patients with ER-positive primary tumors. Consequently, there is an unmet need for novel prognostic and treatment predictive markers, as well as treatment strategies [30]. Our data indicate that low levels of systemic NK cells may be a novel prognostic marker for improved OS in patients with ER-negative MBC, but these results need to be confirmed in a larger patient cohort.

We were previously the first to report that systemic Mo-MDSCs (CD14^+^HLA-DR^low/−^Co-receptor^low/−^ cells), but not all CD14^+^ monocytes, are enriched in the peripheral blood of MBC patients as compared to healthy donors and that Mo-MDSCs associate with clinical parameters and outcome [22,23]. We also recently showed that Mo-MDSCs, but not all CD14^+^ monocytes, associate with ER-negative primary tumors, de novo MBC and worse outcome [23]. Monocyte counts have previously been associated with CTC levels in MBC patients [25] and high levels of CD14^+^HLA-DR^+^ monocytes associated with improved breast-cancer-specific survival in MBC patients treated with high-dose chemotherapy [27]. Furthermore, MBC patients tend to have higher levels of systemic CD16^+^ monocyte subpopulations compared to healthy donors [22], and a previous study has indicated that CD14^+^CD16^+^ monocytes (comprising both intermediate and non-classical monocytes) associate inversely with tumor size in patients with invasive ductal carcinoma, especially at early stages [32]. When analyzing the specific monocyte sub-populations, we observed significant associations between high levels of systemic non-classical monocytes (CD14^+^CD16^++^) and improved OS in patients with ER-positive primary tumors and improved PFS in patients with ER-negative primary tumors. In contrast, classical- (CD14^++^CD16^−^) and intermediate (CD14^++^CD16^+/++^) monocytes did not associate with survival. Yet, classical monocytes associated significantly with hormone-receptor status, whereas high levels of intermediate monocytes associated significantly with more CTCs and worse PFS. The association between intermediate monocytes and PFS warrants further studies as it may open up specific treatment strategies for this patient group. To our knowledge, our study is the first in MBC patients to separate CD16^+^ monocytes into intermediate and non-classical subsets and indicates that distinct monocyte subpopulations associate with specific clinical parameters. Finally, when analyzing peripheral blood DC populations, low levels of MDC1, but not MDC2 or PDC, associated with visceral metastases and a borderline association was observed for high levels of MDC1 and improved PFS in patients with ER-positive primary tumors. Our data are partially in line with a study indicating that high levels of MDC, as well as PDC, associate with improved breast-cancer-specific survival in MBC patients treated with high-dose chemotherapy [27]. MDC1 are potent cross-presenters of antigens to activate Th1 and CD8^+^ T lymphocytes and may associate with an improved anti-tumor immune response [41]. However, DCs are generally present at very low levels in the peripheral blood, urging for cautious interpretations of these findings.

MBC is a challenging disease and currently there is no curable treatment in most cases. Stratification of patients according to the levels of circulating immune cell populations has potential to improve prognostication in this patient group and also provide more knowledge about in which patients immune therapy can have a role. So far, immune therapy is only recommended in patients with triple-negative MBC where atezolizumab and pembrolizumab recently has gained FDA approval [42,43] to be used in combination with chemotherapy. However, studies have indicated that immune therapy can also play a role in other forms of MBC but in what subgroups of patients it is most efficient is currently not known. Recently circulating CD8^+^ and CD4^+^ cells were suggested as treatment-predicting factors in patients with hormone-receptor-positive MBC treated with the CDK-inhibitor palbociclib and pembrolizumab [44]. In the present study, we aimed to look at Ficoll-enriched PBMC populations and excluded contaminating granulocytes such as neutrophils and granulocytic-MDSCs (G-MDSCs) while gating. In a previous study, we found that G-MDSCs are enriched in the PBMC fraction after Ficoll enrichment, and that these G-MDSCs do not correlate with any of the investigated clinicopathological variables or disease severity [45]. Although our findings are limited by the relatively small patient cohort, this is to our knowledge the first study to analyze a broad range of lymphoid and myeloid immune cell populations within the same patient and their potential association with clinical parameters and survival. Moreover, the patients were included ahead of the start of systemic therapy and we employed fresh blood samples (not cryopreserved). Hence, this warrants future investigation in a larger patient population to further validate the prognostic relevance of circulating immune cell populations independently of other factors.

## 5. Conclusions

In this study, we show that high levels of systemic CD8^+^ CTLs correlate with improved PFS and OS in patients with MBC. Similarly, high levels of non-classical monocytes associated with improved OS in patients with ER-positive primary tumors. We propose that the baseline (pre-treatment) levels of systemic immune cell populations, such as CD8^+^ CTLs and non-classical monocytes, may be used to predict clinical outcome in MBC patients and that an improved understanding of the systemic immune response is highly relevant to improve MBC therapy.

## Figures and Tables

**Figure 1 cells-11-01639-f001:**
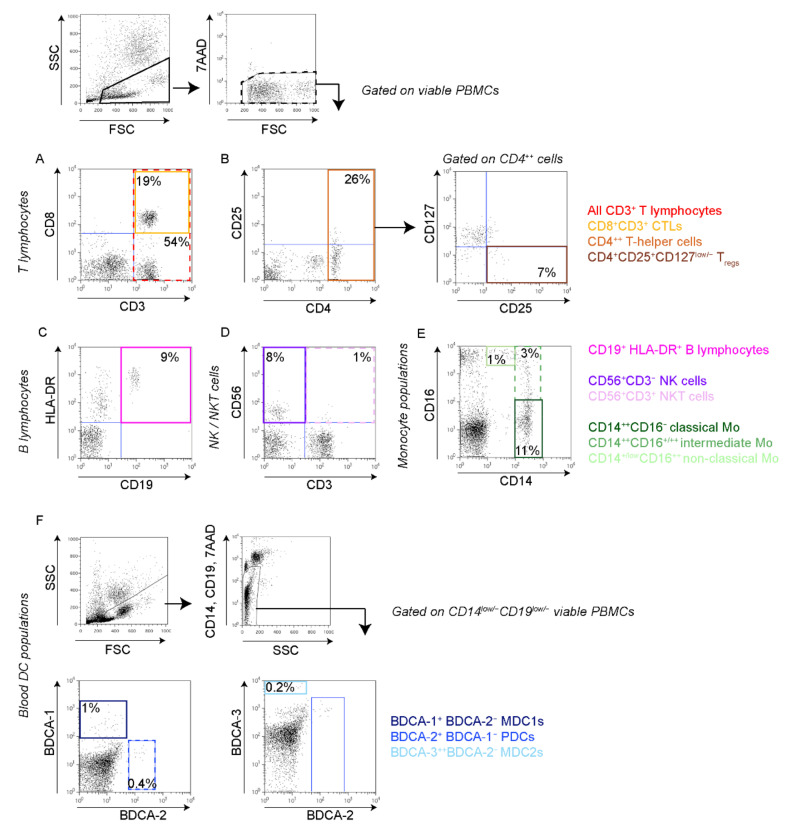
Gating strategies of circulating immune cell populations. Peripheral blood mononuclear cells (PBMCs) from MBC patients were collected and immediately stained for flow cytometry using indicated antibodies. All analyses were performed on gated viable (7AAD-negative) PBMCs (upper panels, black boxes). (**A**,**B**) T lymphocyte populations. (**A**) All CD3^+^ T lymphocytes (red dashed box) and CD8^+^CD3^+^ cytotoxic T lymphocytes (CTLs; yellow box). (**B**) CD4^++^ T helper cells (Th; orange box, **left**) and CD4^++^CD25^+^CD127^low/−^ regulatory T cells (T_regs_; brown box, **right**). (**C).** CD19^+^HLA-DR^+^ B lymphocytes (pink solid box). (**D)** CD56^+^CD3^−^ natural killer (NK) cells (purple solid box) and CD56^+^CD3^+^ natural killer T (NKT) cells (pink dashed box). (**E**) Monocyte populations; CD14^++^CD16^−^ classical monocytes (dark green solid box), CD14^++^CD16^+/++^ intermediate monocytes (green dashed box) and CD14^+/low^CD16^++^ non-classical monocytes (light green solid box). (**F)** Blood dendritic cell (DC) populations, gated on CD14^+/low^CD19^low/−^ viable PBMCs (upper two panels); BDCA-1^+^ MDC1 (dark blue solid box, **left**), BDCA-2^+^ PDCs (blue dashed box, **left**) and BDCA-3^+^ MDC2 (light blue solid box, **right**). All numbers represent percentage in gate.

**Figure 2 cells-11-01639-f002:**
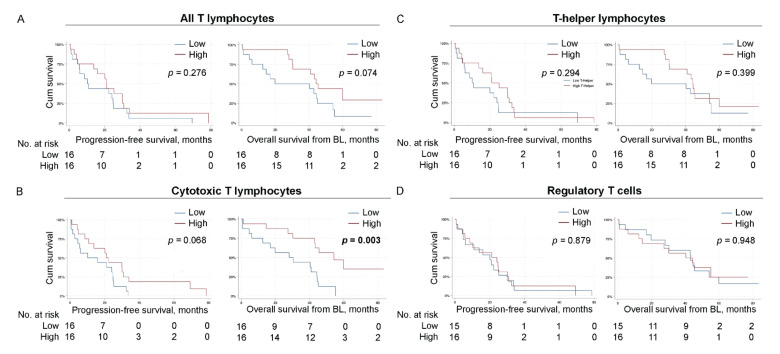
High levels of circulating CTLs associate with improved overall survival. Kaplan–Meier curves of progression-free survival (PFS; **left**) and overall survival (OS) from baseline (**right**) according to the levels of indicated lymphocyte population in all MBC patients. (**A**) all CD3^+^ T lymphocytes. (**B**) CD8^+^ cytotoxic T lymphocytes. (**C**) CD4^+^ T-helper lymphocytes. (**D**) CD4^+^CD25^+^CD127 ^low/−^ T_regs_ of CD4^+^ cells. N = 32 for all populations except for T_regs_ (N = 31). Statistics by log-rank test, *p*-values < 0.05 highlighted in bold.

**Figure 3 cells-11-01639-f003:**
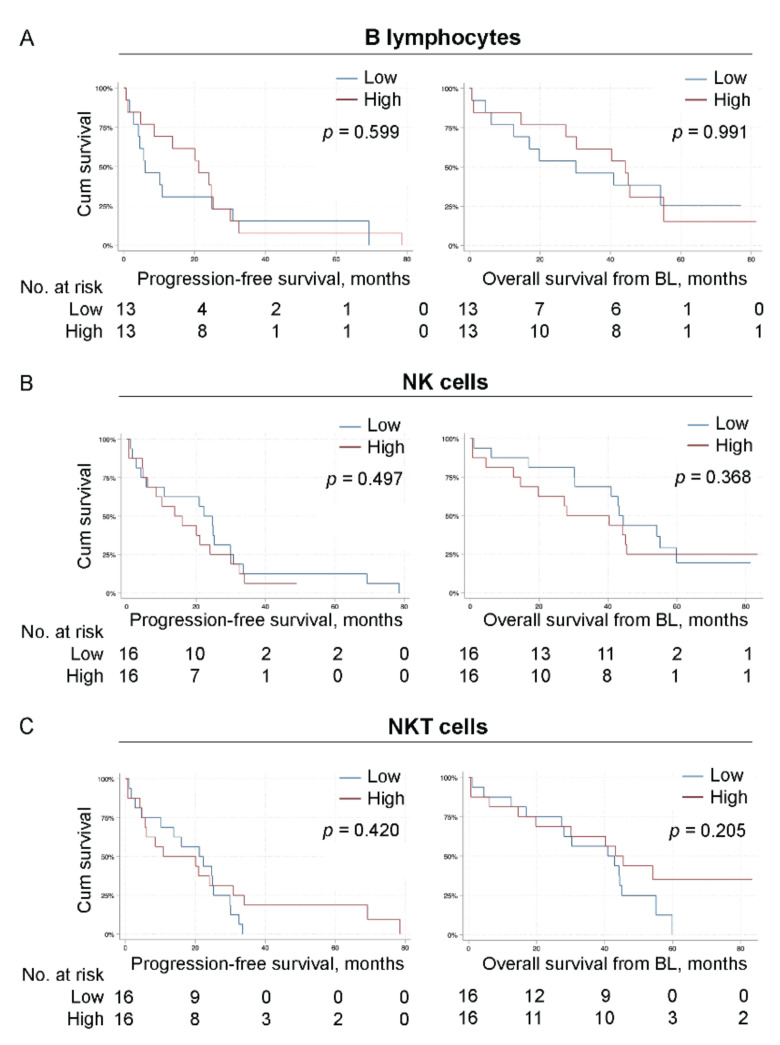
The levels of B lymphocytes, NK cells and NKT cells do not associate with overall survival. Kaplan–Meier curves of progression-free survival (PFS; **left**) and overall survival (OS) from baseline (**right**) according to the levels of indicated lymphocyte population in all MBC patients. (**A**) CD19^+^ B lymphocytes. (**B**) CD56^+^CD3^−^ NK cells. (**C**) CD56^+^CD3^+^ NKT cells. *N* = 32 for all populations except for B lymphocytes (*N* = 26). Statistics by log-rank test with *p*-values indicated.

**Figure 4 cells-11-01639-f004:**
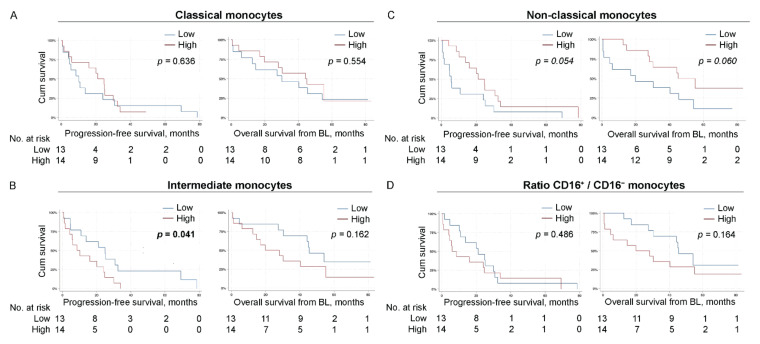
Low levels of intermediate monocytes associate with improved progression-free survival. Kaplan–Meier curves of progression-free survival (PFS; **left**) and overall survival (OS) from baseline (**right**) according to the levels of indicated monocyte subpopulation in all MBC patients. (**A**) CD14^++^CD16^−^ classical monocytes. (**B**) CD14^++^CD16^+/++^ intermediate monocytes. (**C**) CD14^+^CD16^++^ non-classical monocytes. (**D**) ratio of % CD16^+^ monocyte populations/% CD16^−^ monocyte populations. N = 27 for all subpopulations. Statistics by log-rank test, *p*-values < 0.05 highlighted in bold.

**Figure 5 cells-11-01639-f005:**
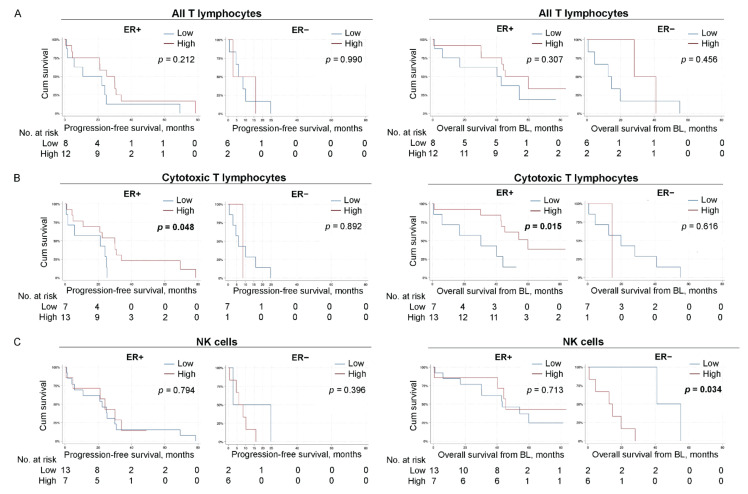
High levels of CTLs associate with improved survival in patients with ER-positive primary tumors. Kaplan–Meier curves of progression-free survival (PFS; **left**) and overall survival (OS) from baseline (**right**) according to the levels of indicated immune cell populations in patients with MBC stratified for primary-tumor ER status. (**A**) all CD3^+^ T lymphocytes. (**B**) CD8^+^ cytotoxic T lymphocytes. (**C**) CD56^+^CD3^−^ NK cells. N = 20 ER-positive and N = 8 ER-negative for all populations. Statistics by log-rank test, *p*-values < 0.05 highlighted in bold.

**Figure 6 cells-11-01639-f006:**
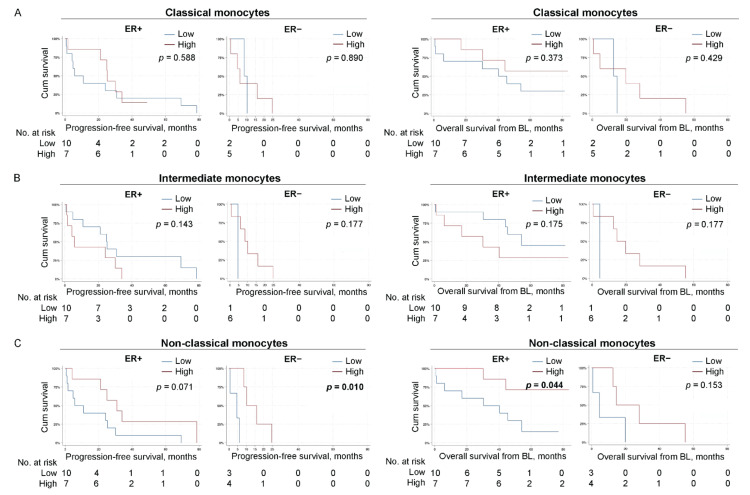
High levels of non-classical monocytes associate with improved PFS in patients with ER-negative primary tumors and with improved OS in patients with ER-positive primary tumors. Kaplan–Meier curves of progression-free survival (PFS; **left**) and overall survival (OS) from baseline (**right**) according to the levels of indicated monocyte subpopulation in patients with MBC stratified for primary tumor ER status. (**A**) CD14^++^CD16^−^ classical monocytes. (**B**) CD14^++^CD16^+/++^ intermediate monocytes. (**C**) CD14^+^CD16^++^ non-classical monocytes. *N* = 17 ER-positive and *N* = 7 ER-negative for all subpopulations. Statistics by log-rank test, *p*-values < 0.05 highlighted in bold.

**Table 1 cells-11-01639-t001:** Summary of *p*-values derived from Pearson’s chi-square test (^a^) or Fisher’s exact test (^b^); used when the expected counts were less than five in at least one cell) when comparing peripheral blood mononuclear cell (PBMC) population levels (dichotomized by using the median as the cut-off) and categorical patient and tumor characteristics. *p*-values < 0.05 were considered significant and are highlighted in bold in the table. See Appendix A for all correlations.

Patient and Tumor Characteristics	CTLs	Tregs	CD8/ CD3	CD4/ CD3	B Cells	NK Cells	Classical Mo	Intermediate Mo	Non-Classical Mo	MDC1
Age	0.077 ^a^	0.200 ^a^	**0.013** ^a^	0.077 ^a^	0.695 ^a^	0.723 ^a^	0.842 ^a^	0.180 ^a^	0.568 ^a^	0.576 ^a^
ECOG	0.109 ^b^	0.493 ^b^	0.666 ^b^	0.666 ^b^	0.181 ^b^	0.859 ^b^	0.134 ^b^	1.000 ^b^	**0.023** ^b^	0.859 ^b^
PT histological subtype	0.754 ^b^	**0.049** ^b^	0.660 ^b^	1.000 ^b^	0.053 ^b^	**0.050** ^b^	0.446 ^b^	0.720 ^b^	0.092 ^b^	**0.007** ^b^
PT NHG	0.336 ^b^	**0.021** ^b^	1.000 ^b^	0.208 ^b^	0.322 ^b^	0.489 ^b^	1.000 ^b^	1.000 ^b^	0.087 ^b^	0.811 ^b^
PT ER	**0.033** ^b^	1.000 ^b^	0.410 ^b^	0.096 ^b^	0.405 ^b^	0.096 ^b^	0.371 ^b^	0.078 ^b^	0.659 ^b^	1.000 ^b^
PT PR	**0.001** ^a^	0.691 ^a^	**0.004** ^a^	**0.00005** ^a^	1.000 ^b^	**0.038** ^a^	**0.006** ^a^	0.292 ^a^	0.827 ^a^	0.462 ^a^
PT HER2	1.000 ^b^	0.338 ^b^	**0.045** ^b^	0.311 ^b^	1.000 ^b^	0.149 ^b^	1.000 ^b^	1.000 ^b^	1.000 ^b^	0.149 ^b^
MET ER	0.429 ^b^	1.000 ^b^	1.000 ^b^	0.464 ^b^	0.478 ^b^	0.464 ^b^	0.478 ^b^	0.435 ^b^	1.000 ^b^	1.000 ^b^
MET PR	1.000 ^b^	0.248 ^b^	0.310 ^a^	**0.047** ^b^	0.391 ^b^	0.696 ^b^	0.099 ^b^	1.000 ^b^	0.387 ^b^	0.228 ^b^
MET HER2	1.000 ^b^	0.593 ^b^	0.225 ^b^	1.000 ^b^	0.553 ^b^	0.238 ^b^	1.000 ^b^	0.229 ^b^	1.000 ^b^	0.565 ^b^
No of metastatic sites (<3 vs. >/=3)	0.072 ^a^	0.347 ^a^	0.719 ^a^	0.280 ^a^	**0.047** ^a^	0.280 ^a^	0.863 ^a^	0.168 ^a^	0.343 ^a^	0.213 ^a^
Visceral metastases	0.072 ^a^	0.886 ^a^	0.719 ^a^	0.280 ^a^	0.420 ^a^	0.719 ^a^	0.695 ^b^	1.000 ^b^	0.236 ^b^	**0.038** ^a^
de novo MBC vs. distant recurrence	1.000 ^b^	0.172 ^b^	**0.043** ^b^	0.333 ^b^	1.000 ^b^	0.333 ^b^	1.000 ^b^	1.000 ^b^	0.595 ^b^	0.654 ^b^
CTC (</≥5)	0.480 ^a^	0.210 ^a^	1.000 ^a^	1.000 ^a^	**0.047** ^a^	0.480 ^a^	0.816 ^a^	**0.034** ^a^	0.310 ^a^	0.366 ^a^
Dead vs. alive	**0.037** ^b^	0.685 ^b^	1.00 ^b^	0.685 ^b^	1.00 ^b^	1.00 ^b^	0.678 ^b^	0.420 ^b^	0.209 ^b^	0.220 ^b^

Abbreviations: *CTLs* cytotoxic T-lymphocytes, *Tregs* regulatory T cells, *NK cells* natural killer cells, *Mo* monocytes, *MDC* myeloid dendritic cell, *ECOG* Eastern Cooperative Oncology Group, *PT* primary tumor, *NHG* Nottingham histological grade, *ER* estrogen receptor, *HER2* human epidermal growth factor receptor 2, *MET* metastases and *CTC* circulating tumor cells.

## Data Availability

The datasets used and analyzed during the current study are available from the corresponding authors upon reasonable request.

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
