# Peer review of "Peripheral Blood Mononuclear Cell Populations Correlate with Outcome in Patients with Metastatic Breast Cancer"

_cells, 2022, doi:10.3390/cells11101639_

Round 1

Reviewer 1 Report

In this manuscript by Larsson et al, the authors investigate pre-treatment fresh blood samples from metastatic breast cancer patients. This is (one of) the first study investigating these types of samples, and might provide information for future diagnostic and treatment purposes. I think the manuscript is very promising and with the below recommendations is suitable for publication in Cells. Please find my comments below - due to comment 2 I marked the manuscript as requiring major revisions.

Major suggestions

  1. While the title is factually correct, the authors have not investigated all leukocytes. By using ficoll paque density gradient separation, and not a whole blood staining, the neutrophils and a large part of the eosinophils have been removed. Therefore, I have two recommendations: 1 – change “leukocytes” to peripheral blood mononuclear cells, as this more clearly reflects what has been investigated. However, upon investigating the gating strategy, there is a significant contamination with granulocytes in the plot. Authors should either discuss this, or set a separate “granulocyte” gate and perform correlation analysis also including this group. If authors do the later, the title can stay as is, otherwise, please discuss how the isolation technique is a limiting factor in the study, preventing assessment of neutrophils, a main source of myeloid-derived suppressor cells and that PBMCs were contaminated with granulocytes.

  2. I do not agree with the gating strategy employed. In whole blood, CD3+CD4-CD8- and CD3+CD4+CD8+ (double negative [DN] and double positive [DP] T cells, respectively), are present. Amongst others, these cells are indicators of recent thymic emigrants, and can be very informative, especially in a disease setting. Therefore, authors should amend their gating strategy – first gate on CD3+CD56- T cells, and subsequently plot CD4 vs CD8, and then define populations as appropriate. Probably, the amount of cells that have been stained is insufficient to detect DP or DN cells, if so, please add this information as a supplementary figure. Nonetheless, pure T cells should be identified by gating CD3+CD56- in order to delineate NKT cells (which can express CD8) [as performed in panel D), and then subdivided by CD4/CD8. Please amend gating strategy, or, if not possible due to the use of separate staining panels, discuss this limitation.

  3. My last point is not a concern, but rather an urging to the authors to use more of their data. Judging by the patient characteristics, also associations between PR status and immune subsets can be made, and number of metastatic sites and immune subsets, vs survival. I can imagine, especially for the number of metastases, that higher circulating immune subsets of, let’s say, CD8+ T cells, can be favourable. Therefore, I would like to invite the authors to perform (some of) these correlations and add this to the manuscript.

Minor suggestions or comments

Line 35 – remove comma after disease

Line 77 – add comma after knowledge

Line 104 – please split this paragraph in “cell isolation” and “flow cytometry”.

Line 110 – please specifiy the EDTA and sucrose concentrations in the wash buffer.

Line 115 – Suppl table 2 comes before suppl table 1, please restructure supplemental information to a chronological order in the text.

Line 118 – please add flow cytometry analysis software and version.

Line 133 – please add a full stop after MBC and start a new sentence.

Section 3.1 – please provide information whether this sub-cohort is representative to the full cohort.

Line 153 – add a comma after cohort.

Line 157 – the gating strategy in a flow cytometry-centred study such as this is not at all supplementary, and should be included in the main text.

Line 164 and onward – please sum up the important (mostly interesting, or significant) results in a table for easier overview for readers and add this to the main text, instead of referring to File 1 that is not included in the main file. This will greatly enhance readability of the paper.

For all figures – please remove “figure x” above all figures. Furthermore, the text near the red/blue lines is illegible. Perhaps just relabel to “Low” and “High” when increasing the font size.

Please either delineate all populations by adding markers (as in panels 1a-c) to all figures, or removing the CD3/CD4/CD8 in panels 1a-c.

Line 292 – thank you for adding this statement about patient numbers, this is very good.

Author Response

In this manuscript by Larsson et al, the authors investigate pre-treatment fresh blood samples from metastatic breast cancer patients. This is (one of) the first study investigating these types of samples, and might provide information for future diagnostic and treatment purposes. I think the manuscript is very promising and with the below recommendations is suitable for publication in Cells. Please find my comments below - due to comment 2 I marked the manuscript as requiring major revisions.

General response: We thank reviewer one for the positive and helpful comments and provide a point by point response regarding the major suggestions below:

Major suggestions

1. While the title is factually correct, the authors have not investigated all leukocytes. By using ficoll paque density gradient separation, and not a whole blood staining, the neutrophils and a large part of the eosinophils have been removed. Therefore, I have two recommendations: 1 – change “leukocytes” to peripheral blood mononuclear cells, as this more clearly reflects what has been investigated. However, upon investigating the gating strategy, there is a significant contamination with granulocytes in the plot. Authors should either discuss this, or set a separate “granulocyte” gate and perform correlation analysis also including this group. If authors do the later, the title can stay as is, otherwise, please discuss how the isolation technique is a limiting factor in the study, preventing assessment of neutrophils, a main source of myeloid-derived suppressor cells and that PBMCs were contaminated with granulocytes.

Response: We thank reviewer one for this observation and agree that the title can be revised to specify the cell populations studied. We have now changed the title to “Peripheral blood mononuclear cell populations correlate with outcome in patients with metastatic breast cancer”.

We agree that the isolation technique using Ficoll comes with limitations. The present study aimed to look at PBMC populations and contaminating granulocytes were excluded in the gating. We have now clarified this in the new Figure 1 as well as in the materials and methods on line 122. Moreover, in an earlier study, we found that granulocytic-MDSCs (G-MDSCs) are enriched in the PBMC fraction after Ficoll enrichment, and that these G-MDSCs do not correlate with any of the investigated clinicopathological variables or disease severity (Mehmeti-Ajradini et al Life Science Alliance 2020, PMID: 32958605). This has now been added to the discussion on lines 582-587.

2. I do not agree with the gating strategy employed. In whole blood, CD3+CD4-CD8- and CD3+CD4+CD8+ (double negative [DN] and double positive [DP] T cells, respectively), are present. Amongst others, these cells are indicators of recent thymic emigrants, and can be very informative, especially in a disease setting. Therefore, authors should amend their gating strategy – first gate on CD3+CD56- T cells, and subsequently plot CD4 vs CD8, and then define populations as appropriate. Probably, the amount of cells that have been stained is insufficient to detect DP or DN cells, if so, please add this information as a supplementary figure. Nonetheless, pure T cells should be identified by gating CD3+CD56- in order to delineate NKT cells (which can express CD8) [as performed in panel D), and then subdivided by CD4/CD8. Please amend gating strategy, or, if not possible due to the use of separate staining panels, discuss this limitation.

Response: Although we fully agree with reviewer one, at the time of the study we did not have access to modern flow cytometers, but a FACS Calibur (allowing for simultaneous analyses of four markers). Thus, more in-depth analyses of T lymphocyte subpopulations were not technically feasible. Consequently, we focused on general markers of immune cell populations and combined staining panels accordingly to maximize the information obtained for each sample. We agree with reviewer one that the use of separate staining panels is a limitation and have expanded the discussion regarding this limitation on lines 487-492 in the discussion.

3. My last point is not a concern, but rather an urging to the authors to use more of their data. Judging by the patient characteristics, also associations between PR status and immune subsets can be made, and number of metastatic sites and immune subsets, vs survival. I can imagine, especially for the number of metastases, that higher circulating immune subsets of, let’s say, CD8+ T cells, can be favourable. Therefore, I would like to invite the authors to perform (some of) these correlations and add this to the manuscript.

Response: We thank the reviewer for this comment. We have now summarized the significant correlations between different PBMC levels and clinicopathological factors in Table 1, including PR status and number of metastases. Furthermore, we have now included survival (dead vs alive) in the new Table 1 (between section 3.1 and 3.2 in the results) where we summarize the P-values when comparing the levels of PBMC subpopulation and categorical patient and tumor characteristics. Here we show that low CTL level is the only PBMC level analyzed that significantly associates with death. The reason why we have stratified the survival analyses based only on ER status is that immune therapy mostly has a role in triple-negative and HER2 positive MBC and it is also for these subtypes TILs are considered most relevant. For ER positive MBC, immunotherapy is not used in clinical practice. Hence, we believe it is important to show that circulating CTLs also show prognostic relevance in ER positive MBC.

Minor suggestions or comments

Line 35 – remove comma after disease

Response: We have now removed the comma.

Line 77 – add comma after knowledge

Response: We have now added the comma.

Line 104 – please split this paragraph in “cell isolation” and “flow cytometry”.

Response: We have now split the paragraph into two.

Line 110 – please specifiy the EDTA and sucrose concentrations in the wash buffer.

Response: The concentrations of EDTA and sucrose was specified on lines 107-106 (now lines 109-110). We have now clarified this on line 110.

Line 115 – Suppl table 2 comes before suppl table 1, please restructure supplemental information to a chronological order in the text.

Response: On line 115 (now line 119) Suppl table 2 comes before suppl table 3 so we are not sure what the reviewer is referring to here. Suppl table 1 is mentioned on line 89.

Line 118 – please add flow cytometry analysis software and version.

Response: We have now included the flow cytometry analysis software and version in the materials and methods on lines 122-123.

Line 133 – please add a full stop after MBC and start a new sentence.

Response: We have now split the sentence into two.

Section 3.1 – please provide information whether this sub-cohort is representative to the full cohort.

Response: The sub-cohort is representative of the full cohort and shows no significant discrepancies with regards to baseline clinicopathological variables (age, ECOG, NHG, subtype, no of metastatic loci, metastasis-free interval, visceral metastasis, number of CTCs and CTC-clusters). This has now been clarified in Section 3.1, on lines 142-146.

Line 153 – add a comma after cohort.

Response: We have now added the comma.

Line 157 – the gating strategy in a flow cytometry-centred study such as this is not at all supplementary, and should be included in the main text.

Response: We have now moved the gating strategy to the main text as Figure 1 and changed the following figure numbers accordingly.

Line 164 and onward – please sum up the important (mostly interesting, or significant) results in a table for easier overview for readers and add this to the main text, instead of referring to File 1 that is not included in the main file. This will greatly enhance readability of the paper.

Response: Please find Table 1 between results section 3.1-3.2, where we have summed up the significant correlations (p<0.05) in a new table to make it more easily accessible.

For all figures – please remove “figure x” above all figures. Furthermore, the text near the red/blue lines is illegible. Perhaps just relabel to “Low” and “High” when increasing the font size.

Response: We have now removed ”figure x” and increased the font size for the ”low” and ”high” labels.

Please either delineate all populations by adding markers (as in panels 1a-c) to all figures, or removing the CD3/CD4/CD8 in panels 1a-c.

Response: We have now removed the markers CD3/CD4/CD8 from Figure 2A-C (previously Figure 1) as well as in Figure 5A-B (previously Figure 4).

Line 292 – thank you for adding this statement about patient numbers, this is very good.

Response: Thank you for pointing this out.

Reviewer 2 Report

Larsson et al.  cells-1671937 " Circulating leukocyte populations correlate with outcome in patients with metastatic breast cancer" is a valuable review paper that shows the baseline levels of systemic immune cell populations and non-classical monocytes may be used to predict clinical outcome in MBC patients and that an improved understanding of the systemic immune response is highly relevant to improve MBC therapy. However, some points were difficult for the reviewer to understand. The reviewer hopes that providing more information (described below) will help improve this study's quality.

About Figures –The panels in all figures are too small to read the text in the panels. Therefore, the reviewer thinks that figures should be resized to make them easier to read it.

About results – All results are done in the same way. Therefore, the reviewer thinks that authors should consider whether a different analysis method yields similar results.

Moreover, to make it easier for readers to understand the authors' point of view, the reviewer thinks that a figure should be added to the manuscript that clearly shows how the present study results can be applied to future treatment.

Author Response

Larsson et al.  cells-1671937 " Circulating leukocyte populations correlate with outcome in patients with metastatic breast cancer" is a valuable review paper that shows the baseline levels of systemic immune cell populations and non-classical monocytes may be used to predict clinical outcome in MBC patients and that an improved understanding of the systemic immune response is highly relevant to improve MBC therapy. However, some points were difficult for the reviewer to understand. The reviewer hopes that providing more information (described below) will help improve this study's quality.

General response: We thank reviewer two for the positive and helpful comments and provide a point by point response regarding the suggestions below:

About Figures –The panels in all figures are too small to read the text in the panels. Therefore, the reviewer thinks that figures should be resized to make them easier to read it.

Response: We thank reviewer 2 for pointing this out and have now increased the font size in all figures to improve the readability.

About results – All results are done in the same way. Therefore, the reviewer thinks that authors should consider whether a different analysis method yields similar results.

Response: We have discussed different statistical methods with our statistician before initiating analyses and were recommended a dichotomized approach for survival analyses using Kaplan-Meier and log rank test to illustrate and compare survival between groups. To compare immune cell levels to patient/tumor characteristics we used the Pearson’s chi-squared test (or if expected counts were lower than 5 in one or more of the cells, Fisher’s exact test was used) which we believe are adequate methods for comparison of different variables in this context. To further investigate a possible association between PBMC levels and death, we have now added these correlations in Table 1 (as well as File 1) and included this information in the results (section 3.2 line 218-219). For the different immune cell populations analyzed, only low levels of CTLs had a significant correlation to death, which supports our previous findings.

In addition, we have now also included multivariable Cox analyses for comparison between levels of CTLs and PFS as well as OS. Similar to the Log-rank analyses in Figure 2 (previously Figure 1), high levels of CTLs displayed non-significant associations to improved PFS in uni- as well as multivariable analyses. For OS, patients with high levels of CTLs significantly associated with improved OS in univariable Cox regression analyses. This significance was, however, lost when adjusting for other prognostic factors (age, ECOG status, NHG, primary tumor histological subtype, metastasis-free interval, number of metastatic sites and site of metastasis). However, the analyses are limited by the relatively small cohort and further analyses in a larger cohort is warranted. We have now included this information in the results (section 3.13) as well as in the discussion on lines 462-464.

Moreover, to make it easier for readers to understand the authors' point of view, the reviewer thinks that a figure should be added to the manuscript that clearly shows how the present study results can be applied to future treatment.

Response: We appreciate this comment and have added a graphical abstract to highlight how the results can be applied.

Reviewer 3 Report

The authors investigate the prognostic significance of peripheral blood leukocyte population in metastatic breast cancer. Although the findings reported are interesting, the analysis suffers because of low numbers and lack of multivariable analysis. The question "have the variables investigated  prognostic significance indepedently of other known clinical and histopatholgic parametes?" is not answered satisfactorily. Unless a multivariable analysis is performed the authors must state the limitation of their study in general (including in the abstract-not simply regarding some parameters).

Also, have they compared the leukocyte populations they investigated with findings in the original tumors? Is there a correlation? If there is that meens that the prognostic information is already registered at the diagnosis, if not that additional immunologic studies (at the detection of metastasis) are needed-they must comment on the issue.

Author Response

General response: We thank reviewer three for the positive and helpful comments and provide a point by point response regarding the suggestions below:

The authors investigate the prognostic significance of peripheral blood leukocyte population in metastatic breast cancer. Although the findings reported are interesting, the analysis suffers because of low numbers and lack of multivariable analysis. The question "have the variables investigated  prognostic significance indepedently of other known clinical and histopatholgic parametes?" is not answered satisfactorily. Unless a multivariable analysis is performed the authors must state the limitation of their study in general (including in the abstract-not simply regarding some parameters).

Response: We have now included multivariable Cox analyses for comparison between levels of CTLs and PFS as well as OS. Similar to the Log-rank analyses in Figure 2 (previously Figure 1), high levels of CTLs displayed non-significant associations to improved PFS in uni- as well as multivariable analyses. For OS, patients with high levels of CTLs significantly associated with improved OS in univariable Cox regression analyses. This significance was, however, lost when adjusting for other prognostic factors (age, ECOG status, NHG, primary tumor histological subtype, metastasis-free interval, number of metastatic sites and site of metastasis). However, the analyses are limited by the relatively small cohort and further analyses in a larger cohort is warranted. We have now included this information in the results (section 3.13) as well as in the discussion on lines 462-464 and also adjusted the abstract (line 28-29)

Also, have they compared the leukocyte populations they investigated with findings in the original tumors? Is there a correlation? If there is that meens that the prognostic information is already registered at the diagnosis, if not that additional immunologic studies (at the detection of metastasis) are needed-they must comment on the issue.

Response: In clinical practice, there is no evaluation of immune cells at initial diagnosis of early breast cancer, hence primary tumors were not evaluated for this. However, in an ongoing study we plan to evaluate different immune cell populations in the primary tumors as well as distant metastasis to investigate correlations and changes in immune cell populations during tumor progression. We believe this is outside the scope of this manuscript but have now clarified this even further in the discussion, lines 482-486.

Round 2

Reviewer 1 Report

I would like to thank the authors for taking my comments and suggestions into consideration and implementing them into the revised manuscript version. I think the manuscript is now suitable for publication, and should be of interest to the scientific community.

Reviewer 2 Report

The second revised paper seems to be improved. However, the reviewer could not check one point in the revised manuscript. The authors answered that we had added a graphical abstract to highlight how the results can be applied to the authors’ responses. There is no graphical abstract in the revised manuscript. The reviewer thinks attaching this graphical abstract to this manuscript is essential.

Reviewer 3 Report

My modifications are accepted. the manuscript is OK